# Drug Use among Nursing Home Residents in Denmark for Drugs Having Pharmacogenomics Based (PGx) Dosing Guidelines: Potential for Preemptive PGx Testing

**DOI:** 10.3390/jpm10030078

**Published:** 2020-07-31

**Authors:** Charlotte Vermehren, Regitze Søgaard Nielsen, Steffen Jørgensen, Anne Mette Drastrup, Niels Westergaard

**Affiliations:** 1Department of Clinical Pharmacology, Bispebjerg Hospital, University of Copenhagen, Bispebjerg Bakke 23, 2400 Copenhagen, Denmark; charlotte.vermehren@regionh.dk (C.V.); regitze.sn@gmail.com (R.S.N.); anne.mette.drastrup@regionh.dk (A.M.D.); 2Department of Pharmacy, Section for Social and Clinical Pharmacy, Faculty of Health and Medical Sciences, University of Copenhagen, Universitetsparken 2, 2100 Copenhagen, Denmark; 3Centre for Engineering and Science, Department of Biomedical Laboratory Science, University College Absalon, Parkvej 190, 4700 Naestved, Denmark; stjo@pha.dk

**Keywords:** nursing homes, polypharmacy, medicine review, pharmacogenomics, cytochrome P450, SLCO1B1, drug–drug interactions, drug–gene interactions, drug use, PGx testing

## Abstract

Background: Polypharmacy is most prevalent among the elderly population and in particular among nursing home residents. The frequency of the use of drugs with pharmacogenomics (PGx)-based dosing guidelines for CYP2D6, CYP2C9, CYP2C19 and SLCO1B1 were measured among nursing home residents in the Capital Region of Denmark as well as drug–drug interactions. The aim was to evaluate the potential of applying PGx-test as a supportive tool in medication reviews. Methods: Drug use among nursing home residents during 2017–2018 in the Capital Region of Copenhagen, for drugs with PGx-based dosing guidelines available through the PharmGKB website, were measured. Drug–drug interactions were scored in severity by using drug interaction checkers. Results: The number of residents using drugs with PGx-based actionable dosing guidelines (AG) were 119 out of 141 residents (84.3%). Of these 119 residents, 87 residents used drugs with AG for CYP2C19, 47 residents for CYP2D6, and 42 residents for SLCO1B1. In addition, 30 residents used two drugs with an AG for CYP2C19, and for CYP2D6, it was only seven residents. The most used drugs with AG were clopidogrel (42), pantoprazole (32), simvastatin (30), metoprolol (25), and citalopram (24). The most frequent drug interactions found with warnings were combinations of proton pump inhibitors and clopidogrel underscoring the potential for phenoconversion. Conclusion: this study clearly showed that the majority of the nursing home residents were exposed to drugs or drug combinations for which there exist PGx-based AG. This indeed supports the notion of accessing and accounting for not only drug–gene but also drug–drug–gene interactions as a supplement to medication review.

## 1. Introduction

According to the World Health Organization (WHO), medication safety in polypharmacy, defined as the administration of multiple drugs at the same time, is one of the key global challenges for medication today [1]. Polypharmacy is an important aspect of drug treatment and becomes inappropriate when the medication risks outweigh benefits for an individual patient due to adverse drug reactions (ADR) and drug–drug interactions (DDI) leading to increased morbidity, mortality and healthcare costs [1,2].

Multimorbidity and thereby polypharmacy are most prevalent among the elderly population and will increase globally in the coming years due to the increasing number of elderly people [3]. Particularly in nursing homes, the incidence of polypharmacy is very high, as shown in the recent SHELTER study [4], among 4023 nursing home residents across Europe. It was found that 49.7% of the residents experienced polypharmacy and 24.3% experienced excessive polypharmacy, findings that were further substantiated in a systematic review on inappropriate medication among nursing home residents [5]. Not surprisingly, both polypharmacy and inappropriate medication were shown to be significant precipitating factors in frequent hospital admissions [6]. In Denmark, around 750,000 people are polypharmacy users, in this case measured as taking more than five drugs, and for people aged ≥ 75 years, 54% of this age group are polypharmacy users [7]. In addition, the use of medication has been shown, paradoxically, to be tightly associated with an increase in the use of drugs following admission to nursing homes [8]. Initiatives both internationally and in Denmark have been taken to support the best clinical management of patients with multi morbidity and polypharmacy [9,10,11], but they are often complicated, requiring multiple specialists being involved in care planning and execution [12].

The Cytochromes P450 (CYP450) drug-metabolizing enzymes are responsible for catalyzing the oxidative biotransformation of 70–80% of all drugs in clinical use to either inactive metabolites or active substances from pro-drugs [13]. Genes encoding for CYP450 isozymes and in particular CYP2D6, CYP2C9 and CYP2C19 have attracted considerable attention as the major target for pharmacogenomics (PGx) testing because they are highly polymorphic and have been shown to affect both drug response and ADR [14,15,16]. The pharmacogenetic impact on the interaction between drug and CYP450 isozymes, referred to as drug–gene interaction (DGI), has been incorporated into clinical actionable dosing guidelines (AG) and non-actionable dosing guidelines (N-AG) for specific DGI and are publicly available through the Pharmacogenomics Knowledgebase (PharmGKB) [17]. Based on drug–gene scores for metabolic activity [18,19,20], DGI are classified into five distinct phenotypes termed “poor metabolisers” (PM), “intermediate metabolisers” (IM), “extensive metabolisers” (EM; normal activity) and “rapid and ultra-rapid metabolisers” (RM and UM) with UM having faster metabolic activity than RM. Single nucleotide polymorphisms (SNP) in the solute carrier organic anion transporter 1B1 (*SLCO1B1*) correlates with an increase in the exposure to statins which can lead to muscle toxicity, a common statin-related ADR occurring in 1–5% of exposed users [21] in a dose-dependent fashion. Since statins are some of the most prescribed drugs [21], many people can be affected by muscle-related ADR. PGx-based AG has been developed for the phenotypes with intermediate or low function of SLCO1B1 [21].

Recently, the term phenoconversion has been introduced as the Achilles’ heel of personalized medicine [22,23] and describes the overlapping of DDI and DGI, referred to as drug–drug–gene interaction (DDGI) [15], which potentially can give rise to “genotype–phenotype” mismatches. This means that a person scored by PGx testing as an UM, EM or IM can in principle be phenoconverted to EM, IM or PM by co-medication.

In a previous exploratory register study [24] we showed that a large fraction of the Danish population, especially the elderly part, are exposed to drugs or drug combinations for which there exist AG and N-AG-related to PGx of CYP2D6 or CYP2C19. The aim of this study was to measure the frequency of the use of drugs with AG and N-AG for CYP2D6, CYP2C9, CYP2C19 as well as SLCO1B1 among nursing home residents in the Capital Region of Denmark in 2017–18 and to evaluate the potential of applying PGx-test for drugs with AG as a supportive tool in medication reviews.

## 2. Materials and Methods

### 2.1. Clinical Dosing Guidelines

The Clinical Pharmacogenetics Implementation Consortium (CPIC) and the Dutch Pharmacogenetics Working Group (DPWG) clinical dosing guidelines for specific gene–drug pairs for CYP2D6, CYP2C9, CYP2C19 and the solute carrier organic anion transporter (SLCO1B1) were used as the source. The guidelines are publicly available through the PharmGKB website (https://www.pharmgkb.org/). Drug use among nursing home residents during 2017–2018, in the municipality of Hoersholm and Hvidovre situated in the Capital Region of Copenhagen, for drugs with dosing guidelines for CYP2D6, CYP2C9, CYP2C19 or SLCO1B1 (PGx targets) were measured according to their ATC codes [25]. Drugs were divided into those with an actionable guideline (AG) defined as at least one clinical recommendation (i.e., dose adjustment, dose monitoring or avoidance of the given drug) different from EM (normal situation) of any of the phenotypes PM, IM or RM. Drugs with non-actionable guidelines (N-AG) were defined as drugs with no clinical recommendation different from EM of any of the phenotypes based on current clinical knowledge. N-AG are only provided by DPWG. Note that the term RM is covering both RM and UM throughout this manuscript. In the case of statins, (SLCO1B1) AG is defined as recommendations different from normal function (N-F) of the carrier of the phenotypes with low- or intermediate function (L-F or I-F).

Drug–drug interactions were scored in severity by using Medscape^®^ drug interaction checker [26] and “interaktionsdatabasen” managed by the Danish Medicine Agency [27]. Warnings are displayed as “monitor closely” or “serious use alternate”.

### 2.2. Ethics

This project was approved by the Danish Data Protection Agency (I-Suite no 05564). According to Danish law, approval by the Danish Council on Ethics was not required for this study, as we only described anonymized drug-use data. Each patient gave informed consent to be enrolled in the project.

### 2.3. Inclusion of Residents with Polypharmacy and Identifying Treatment with Drugs with AG

The general practitioners (GPs) and the staff of the nursing homes identified and included residents, who met the inclusion criteria; age ≥ 65 years, in treatment with ≥5 drugs. The resident gives informed consent, and the GPs accept to participate in the medication review process. Residents receiving terminal care package were excluded. The included residents were screened by using the nursing home’s electronic journal system (CURA) [28] for receiving prescriptions of drugs, with guidelines for CYP2D6, CYP2C9, CYP2C19 and SLCO1B1. The current medications represented the actual medication list of each resident. Both “as needed” and regular prescriptions were registered.

## 3. Results

Table 1 shows the demographic characteristics of the nursing home residents, the mean number of drugs taken and number of prescriptions of drugs with PGx-based dosing guidelines. As shown 48.2% of the residents belong to the age group of 85–98 years and the female gender accounted for 52–60% of the residents across all age groups. The mean number of total drug use was 12.0 and did not change to any extent across age groups. The number of prescriptions (Table 1) of drugs with AG and N-AG for the PGx-targets investigated for showed that out of 310 prescriptions, 232 had drugs with AG corresponding to 75% of the prescriptions. Prescriptions of drugs with AG compared to N-AG were most pronounced for the PGx targets CYP2C19, CYP2C9 and SLCO1B1 whereas for CYP2D6 the opposite was the case.

The residents received in total 234 different drug products of which 196 were prescribed drugs and 38 were “over the counter” (OTC) products. The OTC products were typically nutrition drinks, vitamins and minerals supplements, eye drops and laxatives. Figure 1 shows the ten most used prescription drugs of which six drugs have PGx-based dosing guidelines of either AG or N-AG. The proton pump inhibitors (PPI) covers pantoprazole, lansoprazole, omeprazole and esomeprazole, and statins include simvastatin and atorvastin (see legend to figure). Paracetamol (acetaminophen), which has no dosing guidelines, is by far the most prescribed drug and was used by 139 out of 141 residents.

The number of residents using drugs for which there exist PGx-based actionable guidelines were 119 out of 141 residents corresponding to 84.3%. Of these 119 residents, 87 residents (73.1%) used drugs for which there exists AG for CYP2C19, 47 residents (39.5%) for CYP2D6, and 42 residents (35.3%) for SLCO1B1 (Table 2). In addition, 30 residents (25.2%) used two drugs with an AG for CYP2C19 whereas for CYP2D6 it was only seven residents (5.9%). It should be noted that the number of residents within each PGx target (Table 2) are additive. However, as can be seen from the table, the sum of residents across the PGx targets far exceeds 119, indicating that several residents used drug combinations with AG for, e.g., CYP2D6 and CYP2C19. There were no statistical differences when age groups were compared horizontally using Chi-squared test. However, there was a trend towards the 85–98 years using fewer SLCO1B1 drugs, i.e., statins.

Table 3 presents the drugs used by residents for which there exist PGx-based dosing guidelines and the number of residents using them. In addition, the phenotypes of which action should be taken according to the guidelines are also shown. The drugs are sorted by ATC codes; PPI (A02BC), glucose lowering drug (A10B), antithrombotic agents (B01A), cardiac therapy (C01), beta-blocking agents (C07), analgesics (opioids N02A including codeine), antipsychotics (N05A), antidepressants (N06A) and finally HMG-CoA reductase inhibitors (C10AA). Of 31 drugs with PGx-based dosing guidelines 18 drugs have AG and 13 drugs have N-AG. Moreover, drugs known to be inhibitors of CYP2D6 and CYP2C19 [29] are also marked. Note that the number of users for the different drugs in Figure 1 and Table 2, Table 3 and Table 4 are not additive, unless it is specifically mentioned in the text, since dispensing to the same person can occur for the different drugs. The most used drug with AG and N-AG were clopidogrel and quetiapine, respectively.

Potential DDI were scored in severity by using Medscape^®^ drug interaction checker [26] and “interaktionsdatabasen” [27]. Interactions related to CYP2C19 and CYP2D6 enzyme activity for the most frequent combinations of drugs with AG and non-AG (Table 4) were scored in order to put potential phenoconversion into perspective. It can be seen that out of 11 warnings identified by Medscape, seven warnings coincided with the warnings from “interaktionsdatabasen” albeit the severity scores were different for the combination of clopidogrel and PPI’s which is the most frequent combinations.

## 4. Discussion

In our previous study [24] we showed, for the first time, that a large fraction of the Danish population is exposed to drugs or drug combinations for which there exists PGx-based dosing guidelines and FDA annotations. In spite of this, the use of PGx-based tests have not gained foothold in daily clinical practice probably because the significance has not been recognized, but also because skepticism has been expressed due to question marks about evidence levels [30,31]. However, the main barriers seem to be awareness from physicians and pharmacists as well as validated decision tools and organizational structures requiring multiple specialists being involved in planning and execution [31,32,33]. In addition, it has so far not been considered how the patients’ test results should be stored and disseminated in the health care system. These barriers pose a major challenge in the use of PGx testing in the clinical setting. In this study, we dig one-step deeper and measure the frequency of the use of drugs with AG and N-AG for both CYP2D6, CYP2C19, CYP2C9, and SLCO1B1 at nursing homes in the Capital Region of Denmark. The purpose is to evaluate the potential of applying PGx-test for drugs with AG as a supplementary tool in medication reviews.

Out of the total number of prescriptions for drugs with both AG and N-AG, 75% includes drugs with AG, which is equivalent to 84.3% of the residents actually using at least one drug with an AG. The use of drugs with AG for CYP2C19 are the most prevalent with regard to both single and multiple drug use. Multiple drug use for drugs with AG across PGx targets was also seen (Table 2). In our previous study, we estimated that 15.7% and 61.0% of the Danish population would have the phenotypes UM, IM or PM for which actions in principle should be taken regarding dose adjustments or avoidance if exposed to drugs with AG for CYP2D6 and CYP2C19. The estimates were based on average Caucasian frequencies of DGI reported for CYP2D6 and CYP2C19 [34]. In this study, this translates into 22 and 86 residents with the corresponding phenotypes for CYP2D6 or CYP2C19. If the DGI for SLCO1B1, CYP2C9 and VKORC1 are taken into consideration as well, it has been shown that 99% of a tested population of 1013 subjects have actionable PGx variant(s) in at least one of the above mentioned genes [31]. The prevalence of actionable PGx variants and use of one or more drugs with AG (N-AG) as outlined above supports the notion that a majority rather than a minority of the residents potentially could benefit from a preemptive PGx test [31].

In comparison to the total drug use at nursing homes, drugs with AG are among the most used drugs by the residents and was only significantly surpassed by paracetamol. These findings are compatible with a recent Danish nationwide pharmacoepidemiological study [35] investigating the use of prescription drugs in the elderly population as well as the SHELTER study mapping polypharmacy in nursing homes across Europe [4]. This underlines that the use of drugs with AG are not only widespread among nursing home residents, but among the elderly population in general [4,35]. The most used drugs with AG and N-AG within each drug class (Table 3) very much reflects our previous findings when looking at the total Danish population in general [24]. However, the prevalence of use of the most used drugs in each drug class in this study is significantly higher in comparison to our previous study [24]. For example, the prevalence of use of citalopram (antidepressant), quetiapine (antipsychotic) and tramadol (opioid) in this study are 17.0%, 17.7% and 8.5%, whereas in our previous study the same numbers were 1.8%, 0.1% and 4.6%, respectively. The findings in this study are in alignment with findings in the SHELTER study [4]. In this context, it should be mentioned that the Capital Region of Demark has prepared a recommendation list compiling the first choice of drugs in the primary sector. The most used drugs in each drug category (Table 3) are found on the recommendation lists.

In this study, only three residents were on warfarin therapy. Extensive evidence has confirmed that DGI for VKORC1 and CYP2C9 configure the major determinants of warfarin-required doses and several studies have shown that application of PGx-testing to determine individual dosing of warfarin is promising [36]. In this context it is worth mentioning that there were 63,775 users of warfarin in Denmark in 2018 [37].

Statins are among the most commonly prescribed drugs and in this study 30% of the residents used statins of which simvastatin was the most used in comparison to atorvastatin. It can be estimated by using genetic frequency data from Bank et al. [38], that of 42 residents using statins (Table 3), 10 and 1 residents would have either intermediate or low function of SLCO1B1, respectively, and potentially be at risk for statin induced myopathies.

The publicly available drug-interaction checkers provided by MedScape [26] and the Danish Medicines Agency [27] were used in this study to score for severity of DDI related to inhibition of CYP2D6 or CYP2C19 activity and for comparison of scores between those two trackers. The highest number of potential DDI were seen for the combinations of clopidogrel and PPI’s. A recent review [39] summarized that omeprazole and esomeprazole have a greater effect on CYP2C19 mediated conversion of clopidogrel to its active metabolite and the effect of clopidogrel on platelet reactivity compared to lansoprazole and pantoprazole. This finding is in alignment with the inhibitory actions of these PPIs reported [29] and reflected by the warning “serious use alternate” by Medscape but not by “interaktionsdatabasen”. For comparison, the number of people who redeemed the combinations of clopidogrel and PPI’s in Denmark during 2018 were found to be more than 35,000 (unpublished results) further substantiating that this interaction is very prevalent and should not be neglected. The combinations of sertraline with either metoprolol, oxycodone or codeine were scored as “monitor closely” by Medscape due to the weak inhibitory action of sertraline on CYP2D6 activity. However, the plasma levels of sertraline is affected by the CYP2C19 activity i.e., the plasma level of sertraline, under normal dosing conditions, would increase for CYP2C19 IM or PM [20]. This could potentially lead to CYP2D6 phenoconversion (DDGI) affecting the pharmacological actions of metoprolol, oxycodone or codeine. Similar considerations can be applied for the other combinations shown in Table 4. If the impact of phenoconversion are taken into consideration the balance between “monitor closely” and “serious use alternate” might change significantly towards the latter, which recently have been substantiated for opioids [40,41] and metoprolol [42]. That phenoconversion constitutes a clinical challenge in polypharmacy patients as exemplified both in this study and in our previous study [24] as well as by others [22,43] further substantiates the need for alignment of drug interaction trackers with regard to severity scoring as well as incorporation of gene scores thereby taking phenoconversion into consideration.

In conclusion, it is the author’s belief that there is an unexplored need to supplement medication review with PGx. This is based on the findings, which clearly show that the majority of the nursing home residents are exposed to drugs or drug combinations for which there exist clinical actionable dosing guidelines provided by CPIC or DPWG as well as FDA annotations related to PGx of CYP2D6, CYP2C9, CYP2C19, and SLCO1B1. This indeed underscores the importance of accessing and accounting for not only DDI and DGI but also DDGI where possible. However, it is recognized that it is a complex process demanding multidisciplinary collaborations to obtain infrastructural capacities for good decision-making processes and that further studies are needed to assess the benefit with preemptive PGx-panel testing as the ultimate goal.

## Figures and Tables

**Figure 1 jpm-10-00078-f001:**
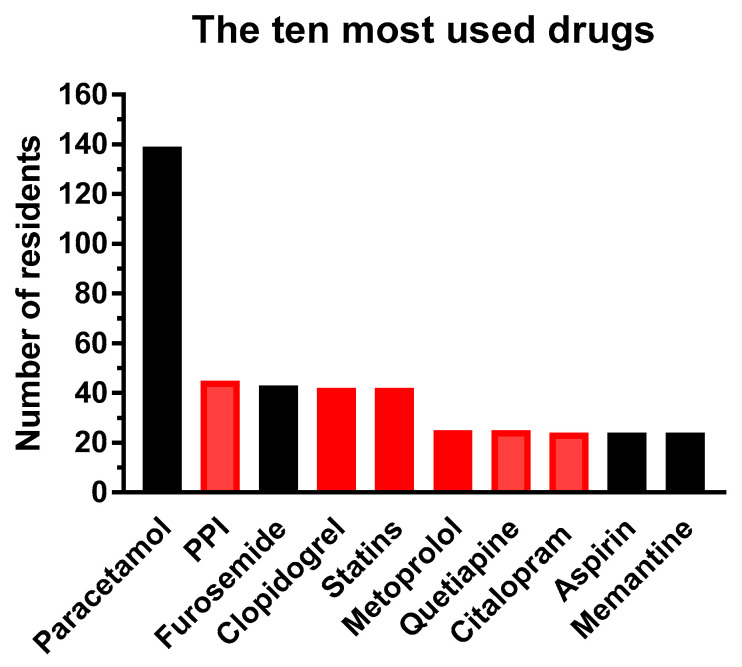
The ten most used drugs and the number of residents treated. Black columns are drugs with no guidelines and red columns are drugs with AG or N-AG. Proton pump inhibitors (PPI) constituted of pantoprazole (32 residents), lansoprazole (seven residents), omeprazole (three residents) all with AG for CYP2C19, and esomeprazole (three residents) with N-AG. Statins cover simvastatin (30 residents) and atorvastin (12 residents) both with AG for SLCO1B1. Metoprolol has AG, quetiapine has N-AG for CYP2D6, and citalopram has AG for CYP2C19.

**Table 1 jpm-10-00078-t001:** Demographic characteristics, drug use and number of prescriptions for pharmacogenomics (PGx) targets.

	Total	Age Groups (Year)		
		65–74	75–84	85–98
No. of residents.	141	22 (15.6)	51 (36.2)	68 (48.2)
No. of female gender	80 (56.7)	13	31	36
Mean no. of drugs	12.0	12.2	13.0	11.5
No. of prescriptions (PGx-target)				
CYP2C19 (AG/N-AG)	133/3	23/1	59/1	51/1
CYP2D6 (AG/N-AG)	54/75	8/16	16/26	30/33
CYP2C9 (AG/N-AG)	3/0	0/0	1/0	2/0
SLCO1B1 (AG/N-AG)	42/0	9/0	23/0	10/0
Total for PGx targets (AG/N-AG)	232/78	40/17	99/27	93/34

Numbers in brackets are percentage expressed relative to total No. of residents. AG: actionable guidelines and N-AG: non-actionable guidelines. The total no of prescriptions in each PGx-target category is the sum of AG + N-AG i.e., total for PGx targets are 232 + 78 = 310 prescriptions.

**Table 2 jpm-10-00078-t002:** Number of residents using drugs with PGx actionable guidelines.

PGx Target	Total Number of Residents. (%)	AGE Groups (Year) Number (%)		
		65–74	75–84	85–98
PGx-drugs (all)	119	20	46	53
CYP2C19 drugs	87 (73.1)	15 (75.0)	38 (82.6)	35 (66.0)
1 drug	50 (42.0)	8 (40.0)	21 (45.7)	21 (39.6)
2 drugs	30 (25.2)	6 (30.0)	11 (23.9)	13 (24.5)
3 drugs	6 (5.0)	1 (5.0)	5 (10.9)	0 (-)
4 drugs	1 (0.8)	0 (-)	0 (-)	1 (1.9)
CYP2D6 drugs	47 (39.5)	6 (30.8)	14 (30.4)	27 (50.9)
1 drug	40 (33.6)	4 (20.0)	12 (26.1)	24 (45.3)
2 drugs	7 (5.9)	2 (10.0)	2 (4.3)	3 (5.7)
CYP2C9 drug	3 (2.5)	0 (0)	1 (2.2)	2 (3.8)
SLCO1B1 drug	42 (35.2)	9 (45.0)	23 (50.0)	10 (18.9)

The percentage shown in brackets are calculated relative to “PGx-drugs all” for the different age group columns. No statistical differences among the groups were found (Chi-squared test).

**Table 3 jpm-10-00078-t003:** No. of residents using drugs with PGx-based dosing guidelines.

Drug	ATC Code	No. of Residents	CYP	RM	EM	IM	PM
Pantoprazole ^2)^	A02BC02	32	CYP2C19	x			
Lansoprazole ^2)^	A02BC03	7	CYP2C19	x			
Omeprazole ^2)^	A02BC01	3	CYP2C19	x			
Esomeprazole ^1)^	A02BC05	3	CYP2C19 *				
Glimepiride	A10BB12	1	CYP2C9 *				
Clopidogrel	B01AC04	42	CYP2C19			x	x
Warfarin	B01AA03	3	CYP2C9	x		x	x
Amiodarone	C01BD01	1	CYP2D6 *				
Metoprolol	C07AB02	25	CYP2D6	x		x	x
Carvedilol	C07AG02	2	CYP2D6 *				
Atenolol	C07AB03	1	CYP2D6 *				
Bisoprolol	C07AB07	1	CYP2D6 *				
Tramadol	N02AX02	12	CYP2D6	x		x	x
Oxycodone	N02AA05	5	CYP2D6 *				
Codeine	R05DA04	8	CYP2D6	x		x	x
Quetiapine	N05AH04	25	CYP2D6 *				
Olanzapine	N05AH03	9	CYP2D6 *				
Risperidone	N05AX08	4	CYP2D6 *				
Aripiprazole	N05AX12	2	CYP2D6				x
Haloperidol ^4)^	N05AD01	1	CYP2D6	x			x
Citalopram ^2,4)^	N06AB04	24	CYP2C19	x			x
Sertraline ^4)^	N06AB06	22	CYP2C19				x
Mirtazapine	N06AX11	17	CYP2D6 *				
Amitriptyline	N06AA09	7	CYP2D6	x		x	x
Amitriptyline	-		CYP2C19			x	x
Venlafaxine	N06AX16	5	CYP2D6	x		x	x
Escitalopram ^2,4)^	N06AB10	2	CYP2C19	x		x	x
Duloxetine ^3)^	N06AX21	2	CYP2D6 *				
Fluoxetine ^3)^	N06AB03	1	CYP2D6 *				
Nortriptyline	N06AA10	1	CYP2D6	x		x	x
				L-F	IM-F	N-F	
Simvastatin	C10AA01	30	SLCO1B1	x	x		
Atorvastin	C10AA05	12	SLCO1B1	x	x		

The table displays all drugs used by the residents for which there exists PGx-based dosing guidelines either actionable (AG) or non-actionable (N-AG; marked with *). Drugs with AG are displayed according to phenotype actions (i.e., dose adjustment, dose monitoring or avoidance of the drug). RM: rapid metabolizer, EM: extensive (normal) metabolizer, IM: intermediate metabolizer and PM: poor metabolizer. Drugs are grouped based on their ATC codes. Amitriptyline have AG for both CYP2C19 and CYP2D6. L-F, IM-F and N-F are low, intermediate or no function of the SCLO1B1 transporter. ^1)^ strong and ^2)^ weak inhibitor of CYP2C19; ^3)^ strong to moderate and ^4)^ weak inhibitor of CYP2D6 according to Flockhart [29].

**Table 4 jpm-10-00078-t004:** Drug combinations with warnings related CYP2C19 and CYP2D6 activity.

		Clopidogrel	Citalopram ^2,4)^	Sertraline ^3)^	Venlafaxine
		42	24	22	5
Omeprazole ^2)^	3	1 ^A)^	1 ^A)^	1	-
Lansoprazole ^2)^	7	5 ^A)^	1	-	-
Pantoprazole ^2)^	32	12 ^A)^	7	3	-
Esomeprazole ^1)^	3	1 ^A)^	-	1	-
Metoprolol	25	-	2 ^B)^	2 ^B)^	1 ^B)^
Oxycodone	5	-	-	2 ^B)^	1 ^B)^
Codeine	8	-	-	2 ^B)^	-

Data are presented as total number of residents using the drugs shown in total (upper and left panel) and in combinations. Drug–drug interaction warnings are scored by using Medscape [26] and Interaktionsdatabasen [27]. Medscape: Yellow; monitor closely; Red: serious use alternate. Interaktionsdatabasen: Underlined: monitor closely. No scores for serious use alternate were found. Warnings shown are related to A) CYP2C19 and B) CYP2D6 enzyme activity. ^1)^ strong and ^2)^ weak inhibitor of CYP2C19; ^3)^ strong to moderate and ^4)^ weak inhibitor of CYP2D6 according to Flockhart [29].

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
