# Peer review of "Drug Use among Nursing Home Residents in Denmark for Drugs Having Pharmacogenomics Based (PGx) Dosing Guidelines: Potential for Preemptive PGx Testing"

_jpm, 2020, doi:10.3390/jpm10030078_

Round 1
Reviewer 1 Report
Dear Authors,
your study is well designed and it is an important study to continue your previous effort for utilizing PGs testing in clinical decision for actionable dosing guidelines. I only have 2 comments:
1- In your previous study "Drug Use in Denmark for Drugs Having Pharmacogenomics (PGx) Based Dosing Guidelines from CPIC or DPWG for CYP2D6 and CYP2C19 Drug–Gene Pairs: Perspectives for Introducing PGx Test to Polypharmacy Patients", you concluded that the numbers of users who redeemed the drugs searched for in this publication does not provide any information about dose, compliance, and clinical effects, as well as duration of treatments. Do you suggest with this current work findings any of the above mentioned and how can you relate to current practice.
2- Please check your English language for minor spelling and English errors.
Thanks.
Author Response
Dear Reviewer,
Thank you very for your comments and suggestions.
The main aim of this particular study was to map the use PGx drugs in nursing homes.
However, we are in the progress (new manuscript) to look more deeply into compliance, dose, side effects and duration for some of the most prevalent PGx drugs and drug combinations used at nursing homes. Unfortunately, for the time being we don't have verified data to add to this manuscript.
The manuscript will be checked for errors
Reviewer 2 Report
The paper titled “Drug use among nursing home residents in Denmark for drugs having pharmacogenomics based (PGx) dosing guidelines: Potential for preemptive PGx testing” by Charlotte Vermehren evaluated the potential of applying PGx-test for drugs have actionable dosing guidelines (AG) as a supplementary tool in medication reviews by analyzing the frequency of the use of drugs having AG for both CYP2D6, CYP2C19, CYP2C9, and SLCO1B1 at nursing homes in the Capital Region of Denmark, and also the need and importance of the PGx related to the drug-genes and drug combinations. The data analysis is reasonable, and the results can well support the conclusion.
I have only one minor concern of Table 1 and subsequent analysis. All the N-AG prescriptions for nursing home residents were pronounced for the PGx targets CYP2C19 and CYP2D6, which are 133 (AG)/3(N-AG) and 54/75, respectively, are there differences between the two groups of residents in terms of drug reactivity?
Minor revision.
Author Response
Thank you very for your comments
The main aim of this particular study was to map the use PGx drugs in nursing homes.
Your question is very relevant and we are in the progress (new manuscript) to look more deeply into these matters such as efficacy/reactivity etc. for the most prevalent PGx drugs and drug combinations used at nursing homes. Unfortunately, for the time being we don't have verified data to add to this manuscript.